# FABP4 Controls Fat Mass Expandability (Adipocyte Size and Number) through Inhibition of CD36/SR-B2 Signalling

**DOI:** 10.3390/ijms24021032

**Published:** 2023-01-05

**Authors:** Emmanuelle Berger, Alain Géloën

**Affiliations:** Laboratoire Ecologie Microbienne (LEM), Unité Mixte de Recherche Centre National de la Recherche Scientifique 5557, Institut National de Recherche pour l’Agriculture, l’Alimentation et l’Environnement 1418, VetAgroSup, Université Lyon 1, Domaine Scientifique de La Doua, 69100 Villeurbanne, France

**Keywords:** adipocyte size, adipogenesis, lipolysis, signalling, fatty acid binding protein 4, fatty acid translocase FAT/CD36 (SR-B2)

## Abstract

Adipose tissue hypertrophy during obesity plays pleiotropic effects on health. Adipose tissue expandability depends on adipocyte size and number. In mature adipocytes, lipid accumulation as triglycerides into droplets is imbalanced by lipid uptake and lipolysis. In previous studies, we showed that adipogenesis induced by oleic acid is signed by size increase and reduction of FAT/CD36 (SR-B2) activity. The present study aims to decipher the mechanisms involved in fat mass regulation by fatty acid/FAT-CD36 signalling. Human adipose stem cells, 3T3-L1, and its 3T3-MBX subclone cell lines were used in 2D cell cultures or co-cultures to monitor in real-time experiments proliferation, differentiation, lipolysis, and/or lipid uptake and activation of FAT/CD36 signalling pathways regulated by oleic acid, during adipogenesis and/or regulation of adipocyte size. Both FABP4 uptake and its induction by fatty acid-mediated FAT/CD36-PPARG gene transcription induce accumulation of intracellular FABP4, which in turn reduces FAT/CD36, and consequently exerts a negative feedback loop on FAT/CD36 signalling in both adipocytes and their progenitors. Both adipocyte size and recruitment of new adipocytes are under the control of FABP4 stores. This study suggests that FABP4 controls fat mass homeostasis.

## 1. Introduction

Adipose tissue is a complex tissue that in obesity induces metabolic disorders such as type II diabetes. Adipose tissue expansion proceeds through [1,2,3,4,5,6] (1) hypertrophy, i.e., adipocyte size (50–70 μm) increase (namely “adiposize”) leading to large (70–120 μm) then very large (120–300 μm) adipocytes associated with insulin resistance, increased lipid mobilization and glucose metabolism, together with (2) hyperplasia, i.e., increased cell numbers. Fatty acid storage capacities as well as adipokine secretions depend on cell size [2,7,8,9,10]. Fat storage in adipocytes is the result of (1) free fatty acid (FFA) uptake and storage as triglycerides (TG) into droplets, (2) FFA release by lipolysis [10,11]; (3) induction of intracellular signaling pathways including fatty acid receptors (mainly translocase FAT/CD36 also named SR-B2), intracellular fatty acid binding proteins (FABPs, mainly FABP4) and glucose uptake (mainly the glucose transporter GLUT4 in mature adipocytes); (4) an adipogenic programme orchestrated by Peroxisome Proliferator Activated Receptor Gamma (PPARG); and (5) mitochondrial fatty acid (FA) de novo synthesis from glucose at least in immature cells [10,11,12,13,14,15]. 

Adipogenesis is induced and regulated by lipogenesis, including de novo lipid synthesis and lipid uptake. In 2D cell cultures in presence of high glucose concentration (4.5 g/L), an improved intracellular lipid accumulation during adipogenesis is obtained in response to oleic acid. Adipogenesis promotes gene transcription of adipogenic markers *FAT/CD36*, *FABP4*, and *CIDEC* [12]. FAT/CD36 is characterized by a high adipogenic potential required for long-chain fatty acid uptake as well as lipolysis [16]. Fatty acid interaction with FAT/CD36 stimulates the transcriptional activity of PPARs, and noticeably PPARG, to induce *CIDEC*, *FABP4* as well as its own gene transcription, thus creating a pro-adipogenic amplification loop [11,16,17,18,19]. However, using a specific cell-permeable inhibitor [20], we found that FAT/CD36 reduces the capacity of TG storage [12]. In adipose tissue, PPARG is required for induction of adipogenesis [21], although a reduction of PPARG is observed in mature adipocytes [22], and in cell cultures, The PPARG gene transcription is repressed after treatment with oleic acid, although its activation by rosiglitazone reduces lipid droplet size [23]. FAT/CD36 mediated activation of PPARG is required for induction of adipogenesis [24,25]. PPARG downregulation could be involved in the regulation of adipocyte size. In another way, fatty acid interaction with FAT/CD36 induces AMPK (AMP-Activated Protein Kinase) activation [26], which inhibits lipolysis through ATGL (adipose triglyceride lipase) and HSL (hormone-sensitive lipase) phosphorylation [27,28,29]. FAT/CD36 deficiency in mature adipocytes is associated with increased basal lipolysis [30]. The lipolytic pathways allow fatty acids to be released together with secretion of FABP4 [31,32], which itself can interact with FAT/CD36 [33], inhibits PPARG and is inversely correlated with PPARG in adipose tissues [34].

The present study aimed to further describe the signalling pathways involved in the differential roles of fatty acid-induced FAT/CD36 signalling in the regulation of fat mass expansion, i.e., adipocyte size and pro-adipogenic processes, using real-time monitoring of 2D cell culture models. Similar to human adipose stem cells (ASCs), in mouse 3T3-L1 cell lineage, differentiation is highly heterogeneous. High discrepancies in droplet size increase after treatment with oleic acid has been observed after 3 days. The fatty acid uptake is a fast event, and real-time image monitoring revealed that lipid exchanges can occur between cells due to high glucose-induced basal lipolysis [23]. This suggests that after fatty acid uptake, which is a fast event, important fatty acid exchanges occur between adipocytes, potentially related to droplet size and/or lipid storage capacities. Our hypothesis suggests that although FAT/CD36 improves the induction of adipogenesis, a reduction of FAT/CD36 and PPARG activities are involved in the regulation of adipocyte size. The present study aimed to decipher the differential roles of these pathways in the regulation of fat mass homeostasis. Using oleic acid induction of droplet size increase and several specific inhibitors in real-time experiments, both induction of FAT/CD36-PPARG mediated gene transcription and FAT/CD36-AMPK inhibition of basal lipolysis were found to promote adipogenesis and adipocyte maturation. Their inhibitions were required to increase adipocyte size. Results show that FAT/CD36 regulation of adipocyte size, together with the recruitment of new adipocytes, are orchestrated by the management of FABP4 accumulation into mature adipocytes.

## 2. Results

### 2.1. FAT/CD36 Is Involved in Lipid Storage but Reduces Lipogenesis

The role of FAT/CD36 in the regulation of adipocyte size was investigated on rat adipose tissue explants. In a previous study, we showed using AdipoRed probe incorporation that oleic acid uptake induced size increase with better efficiency in high (HG, 4.5 g/L) versus low glucose media (LG, 1 g/L) [23]. In the present study, rat explants were treated for 48 h in either LG or HG media and/or optimized concentration of FAT/CD36 inhibitor AP5258 (Figure 1A). Triglyceride (TG) contents were then assessed using the fluorescent probe AdipoRed incorporation in 40 min. We found that in HG media AdipoRed incorporation was higher in HG than in LG media, suggesting that fatty acid de novo synthesis from glucose was detected in these experiments. AP5258 inhibited this effect, suggesting that FAT/CD36 plays a major role in the regulation of TG synthesis in adipose tissue.

Whether fatty acid/FAT/CD36 signalling pathways were involved in the regulation of cell size was monitored in 2D cell cultures (Figure 1B). The 3T3-L1 adipocytes were treated with a low dose of oleic acid in HG culture media in presence of several inhibitors at optimized concentrations over 3 days. Fatty acid uptake is a fast event and this protocol was previously optimized in order to avoid cell detachment or toxicity, given the time for synthesis of the proteins required for lipid accumulation and droplet size increase. This means also that in such experiments, important cell reorganization could be observed, such as basal lipolysis and fatty acid exchange between adipocytes [23]. As previously observed, insulin reduced the effect of OA on droplet size increase. Inhibition of PPARG (by GW9662) or PPARA (by GW6471) did not have any significant effect, suggesting that their regulation by FAT/CD36 was not involved in these conditions. However, inhibition of FAT/CD36 (by AP5258), together with those of ATGL (by ATGListatin) and intracellular FABP4 (by FABP4i), were related to an increase of droplet size in 3T3-L1 cells. This result shows that ATGL-dependent lypolysis, FAT/CD36, and FABP4 play a major role in the regulation of cell size. 

### 2.2. Transcriptional Regulations by Fatty Acid in Absence of FAT/CD36: Emerging Role of FABP4

The regulation of adipocyte size proceeds through the transcriptional induction of genes involved in fatty acid storage. We previously found that oleic acid is a major activator of FAT/CD36—PPARG signalling pathway [12,23]. Whether FAT/CD36 inactivation regulates gene transcription was analyzed using comparative analyzes of gene datasets. The gene dataset regulated by fatty acids was compared to the gene datasets regulated by FAT/CD36 knock-out. The method consists in comparing significant enrichment of genes (genome as reference) [12,23]. Although the datasets are obtained from experiments performed in different conditions and on different kinds of samples, previous studies have shown that significant enrichments in gene datasets allow the identification of signalling pathway crosstalks. A pairwise comparison of gene datasets regulated by either fatty acids or in FAT/CD36 knock-out mice was performed. Both gene datasets were compared to human gene datasets regulated by a series of either extracellular signals, intracellular pathways, or transcriptional regulators. In another way, their relevance to adipose phenotype, i.e., to signalling pathways, is significantly over-represented in either adipose stem cells (ASCs), in vitro differentiated adipocytes (dA), or adipose tissue (AT) datasets has been analyzed (Appendix A). Only signalling pathways significantly over-represented in both fatty acid and FAT/CD36 knock-out gene datasets are presented in Figure 2. 

Eight extracellular signalling pathways are commonly regulatable by FAT/CD36 inhibition and free fatty acid treatment, including glucose, insulin, and several pro-inflammatory signals. Sixteen intracellular signalling pathways are commonly involved in fatty acid signalling without FAT/CD36, they are related to oxidative stress, such as Glucose oxydase, Janus kinase 2 (JAK2), and NADPH oxidase 1 (NOX1). Eight transcriptional regulators, including intracellular FABP4 (FABP4i), were also found in this analysis. Importantly, PPARs did not emerge from this study as expected since they are activable by fatty acids through FAT/CD36 signalling. Intracellular FABP4 is involved in fatty acid-induced signalling in absence of FAT/CD36, in pathways over-represented in either ASCs, in in vitro differentiated adipocytes (dA) as well as in adipose tissue (AT). These data suggest that FABP4 plays a major role in adipose cells with low FAT/CD36 activity. FABP4 was also identified as an external signal with significant transcriptional activities commonly modulated in FAT/CD36 knock-out cells but without a significant link with fatty acid activity (Appendix A). Moreover, both extracellular and intracellular activities of FABP4 present a significant similarity in gene transcription activity (Appendix A). The signalling pathways commonly enriched in extracellular and intracellular FABP4, fatty acids and CD36 knock-out datasets are reported in Figure 2. They include MAPK1/ErK1/2 and JAK2 pathways (Figure 2, central panel). Commonly with fatty acids, FABP4 may also regulate genes regulated by transcription factors CEBPG and Z, LXR, and the USF. These results suggest that FABP4 regulates gene transcription through several signalling pathways independently of FAT/CD36. Thus the role of FABP4 in the regulation of fatty acid signalling and fat mass was further investigated. 

### 2.3. Intracellular FABP4 Regulates Oleic Acid-Induced PPARG Activity in Adipocytes

At the intracellular level, FABP4 plays a pivotal role in the regulation of lipid storage, through its implication in the processes of lipolysis, lipid uptake, and PPARG-mediated gene transcription. FABP4 intracellular accumulation could result from (1) de novo synthesis after fatty acid induction of its gene transcription through FAT/CD36-PPARG signalling and (2) inhibition of lipolysis. The role of intracellular FABP4 in the regulation of size was then assessed in 3T3-MBX adipocytes (Figure 3).

In fully differentiated 3T3-MBX adipocytes, in the presence of OA, FABP4 inhibition hindered the OA-induced increase of droplet size (Figure 4B–D). This result suggests again that intracellular FABP4 is required at least during fatty acid-induced adipogenesis.

In the next step, the role of FABP4 in fatty acid mediated FAT/CD36 signalling was further explored. The recombinant FABP4 protein (FABP4r) and the cell-permeable inhibitor (FABP4i) were used to study its extracellular and intracellular activities, respectively. Activators of PPARG (rosiglitazone) and AMPK (AICAR) were used as controls for FAT/CD36 signalling related to gene transcription and basal lipolysis, respectively. The regulation of FAT/CD36 activation allowing fatty acid signalling was assessed using a specific APC-coupled antibody directed against the extracellular part of FAT/CD36. In 3T3-L1 adipocytes, the extracellular distribution of FAT/CD36 (related to its activable form) was inversely related to lipid droplet size (Figure 4A), suggesting that extracellular FAT/CD36 is downregulated when adipocyte size increases.

The 3T3-L1 adipocytes were treated with oleic acid in order to increase lipid content and maturation. The presence of FAT/CD36 inhibitor promoted lipid droplet size increase without affecting the frequency of “mature” adipocytes (i.e., droplets up to 50 μm diameter). This suggests that FAT/CD36 was no longer involved in the increase of size after maturation (Figure 4B). Interestingly, inhibition of intracellular FABP4 induced an increase of extracellular FAT/CD36 without influencing lipid droplet size but by reducing the frequency of “mature” adipocytes (Figure 4B). These results suggest that FABP4 is involved in the regulation of the activable form of FAT/CD36 during the pro-adipogenic process and is involved in the regulation of adipocyte size. Taken together, these results support the hypothesis that concomitant reduction of FAT/CD36 active form and intracellular accumulation of FABP4 is required to obtain mature adipocytes. 

In order to mimic fatty acid induction of adipocyte maturation, 3T3-MBX adipocytes were treated over 3 days with a low dose of oleic acid (5 μM) in the first step. In the second step, the adipocytes were treated over 48 h with either recombinant FABP4r, cell-permeable inhibitor of FABP4 (FABP4i), PPARG activator (rosiglitazone), or AMPK activator (AICAR) (Figure 4E). As expected, *FABP4*, *FAT/CD36*, and *CIDEC*, another adipogenic marker used as control, were induced by PPARG activation. Both FABP4r addition and inhibition of intracellular FABP4 downregulated FAT/CD36 and CIDEC. In the third step, their lipid storage capacity was assessed by treatment with oleic acid 10 μM over 24 h (Figure 4F). AMPK activation potentiated the effect of oleic acid and promoted droplet size increase, suggesting involvement of its protective effect against basal lipolysis. Both PPARG activation and inhibition of intracellular FABP4 reduced lipid storage capacity. Intracellular FABP4 was required to increase droplet size, since it is involved in *CIDEC* induction and/or maintenance, although FABP4 reduced FAT/CD36 activity. FABP4 and PPARG activities were thus inversely regulated. The extracellular activity of FABP4 was further explored.

### 2.4. Extracellular FABP4 Inhibits the Recruitment of Adipogenic Precursors by Oleic Acid-Induced FAT/CD36 Signalling

FABP4 is secreted by adipocytes during lipolysis. Human adipocytes were treated with oleic acid over 48 h in order to increase droplet size (Figure 5A). Fatty acid uptake is a fast event. Forty-eight hours after treatment with OA, an accumulation of FABP4 in culture media was detected. This observation suggests that basal lipolysis occurred in high glucose culture media. 

In a previous study, we found that basal lipolysis can be detected in high glucose culture media of highly differentiated adipocytes with heterogeneous sizes [23]. We used fully differentiated 3T3-MBX adipocytes obtained by pre-treatment with OA over 3 days (Figure 5B,C) in order to study in co-cultures how basal lipolysis and FABP4 release can modulate the recruitment of new adipocytes using 3T3-L1 fibroblasts. The extracellular FABP4 activity was assessed in 3T3-L1 fibroblasts co-cultured with fully differentiated 3T3-MBX adipocytes (Figure 5D). After 3 days of co-culture, in 3T3L-1 cells the lipid content was higher in high (4.5 g/L) versus low (1 g/L) glucose culture media, due to the uptake of fatty acids released by lipolytic 3T3-MBX adipocytes. However, at the transcriptional level, *FAT/CD36*, *CIDEC*, and *G0S2* were reduced in high versus low glucose co-culture media, suggesting a reduction of the adipogenic process (Figure 5E). 

The role of lipolysis products i.e., fatty acids (represented by oleic acid) and FABP4, were tested on human ASCs in real-time experiments (Figure 6). ASCs exposed to oleic acid during the proliferative phase increased their capacity to differentiate in a dose-dependent manner, the non-toxic dose was defined for a Kd 0.5 μM (Figure 6A). Oleic acid induction of adipogenesis promoted the increase of droplet size according to time (Figure 6B). Recombinant FABP4 did not modulate proliferation, differentiation (Figure 6C), or OA uptake (Figure 6D). Similar results were observed in adipocytes (Appendix A). However, OA complexation to FABP4 counteracted the differentiation process induced by OA complexed to albumin (Figure 6E), without significant effect on lipid accumulation (Figure 6F) but by altering lipid droplet formation (Figure 6G).

Whether extracellular FABP4 stabilization of extracellular addressing of FAT/CD36 correlates with its ability to reduce FAT/CD36 activity was further explored. Fluorescently labeled FABP4 interacts with high affinity to 3T3-MBX cell membranes, with apparent co-localization with polymerized beta-actin cytoskeleton detected with phalloïdin (Figure 7A,B). FABP4 restored extracellular FAT/CD36 addressing in high glucose media in short-term experiments but reduced extracellular FAT/CD36 addressing after long-term exposure (Figure 7C), without affecting the lipogenic activity of oleic acid nor the protective effect of insulin (Appendix A). This result suggests that FABP4 promotes FAT/CD36 activation but that its accumulation reduces FAT/CD36 expression. 

The results are consistent with the results obtained on the bio-informatic gene dataset analyses showing common transcriptional activities of both extracellular and intracellular FABP4 also commonly regulated by fatty acids and in FAT/CD36 knock-out (Figure 2). Among them, MAPK1/2 and JAK2 were identified. Thirteen transcription regulators datasets are commonly over-represented in both extracellular and intracellular FABP4 activities, 11 of them are also involved in JAK2 signalling, and among them, 6 are also involved in LKB1/AMPK1 signalling (Appendix A).

Taken together these results suggest that FABP4 could be internalized to promote the signalling pathways regulated by intracellular accumulation of FABP4 in adipocytes. Since it was not significantly detected in adipogenic precursors, FABP4 transcriptional activity in presence of oleic acid was analyzed on 3T3-L1 fibroblasts and compared to PPARG activation with rosiglitazone (Figure 8A). *CD36*, *CIDEC*, *G0S2*, and *FABP4* are commonly induced by OA and PPARG, they are inhibited by OA-FABP4 as well as by OA when FAT/CD36 is inhibited. Both *FABP4r* and *FAT/CD36* inhibition were linked to the reduction of active FAT/CD36 form (Figure 8B). Taken together these results suggest that FABP4 antagonizes OA-FAT/CD36 signalling pathways in both adipocytes and their precursors.

## 3. Discussion

In obesity fat mass expandability is a result of multifactorial processes including dysregulation of adipocyte size (hypertrophy) and recruitment of new adipocytes from adipose stem cells (hyperplasia). In normal adipose tissue, in non obese individuals, adipocyte cell size distribution is highly homogeneous i.e., in a range of 50 to approx. 100 μm [5,35,36,37], suggesting the existence of a limiting factor to the size increase. In the present study, we identified the pivotal role of FABP4 in the regulation of this homeostasis. The fatty acid-induced activity of FAT/CD36 was highly regulated by FABP4. FABP4 controls adipocyte size as well as the recruitment of new adipocytes.

Using 2D cell cultures we previously determined that lipid droplets modulate cell adhesion force, which can be monitored in real-time experiments using an xCelligence sensor. Fatty acid uptake is a fast event. Pre-treatment of adipocyte precursors by oleic acid promotes induction of adipogenesis (Figure 6A), OA increases the rate of differentiation, and adherent mature adipocytes could be obtained 3 days after uptake of oleic acid (10 μM in HG glucose, 4.5 g/L, culture media) [12,23]. Both induction of adipogenesis and droplet size increase are accompanied by OA/PPARG-induced expression of *CIDEC* and *G0S2*, involved in droplet formation, *FAT/CD36* and *FABP4* adipogenic markers (Figure 4A and Figure 8A).

FAT/CD36 expression is induced by PPARs activation during adipogenesis [16]. It is highly induced in both adipocytes and their precursors through FAT/CD36-PPARG signalling (Figure 4E and Figure 8A) [12,23]. FAT/CD36 interacts with FFA when their extracellular concentration is low but is not required for high lipid uptake [38]. However, we found that FAT/CD36 is required to retain the capacity of lipogenesis induced by high glucose in adipose tissue (Figure 1A). FAT/CD36 level of expression together with its trafficking are determinants of lipolysis [6,39]. OA treatment activates FAT/CD36 in adipocyte precursors (Figure 8B) although this active form is inversely correlated with droplet size increase (Figure 4A). These observations correlate with the increase of adipocyte size observed in PPARG knock-out mice [40], its low level of expression in mature primary human adipocytes [22], and the inhibitory effect of rosiglitazone on induction of size increase (Figure 4F). PPARG and CEBPA over-expression in obese AT [41] are probably a result of increased recruitment of new adipocytes (hyperplasia) rather than of adipocyte size increase since PPARG downregulation is correlated with fat mass reduction and adipocyte hypertrophy [40]. Furthermore, PPARG overexpression and activation in mature adipocytes do not influence adipose cell hypertrophy [42]. Therefore reduction of PPARG activity during the process of lipid uptake allows an increase in size.

During adipogenesis, FABP4 expression is induced by PPARG, and its intracellular activity is required for adipogenesis (Figure 3 and Figure 4F) and lipolysis (Figure 1B). During lipolysis, fatty acids are complexed to FABP4, which represents approximately 1% of all soluble proteins in adipose tissue [43,44]. We found that potential extracellular FABP4 uptake could occur in adipogenic precursors (Figure 7). Intracellular interaction of FABP4 with cytokeratin, a cytosquelettal associated protein [45], and exosome-like export of FABP4 from adipocytes also support the hypothesis of FABP4 uptake by ASCs [46]. FABP4 is required for both fatty acid uptake and release through lipolysis but also in fatty acid signalling to regulate PPARG transcriptional activity [34,43]. Our study suggests that the increase of FABP4 stores due to uptake in adipose progenitors (Figure 6E–G, Figure 7 and Figure 8) or in adipocytes (Figure 4E,F) were correlated with the reduction of the active form of FAT/CD36. FABP4 abundance is inversely correlated with that of PPARG in human adipose tissues and FABP4 induces proteasomal degradation of PPARG [34]. FAT/CD36 activation by FFA activates AMPK, which in turn inhibits lipolysis through the phosphorylation of ATGL [29]. 

Thus the relative availability of FABP4 on either fatty acid uptake, lipolysis, and modulation of PPARG gene transcription may result from the regulation of its interactome, including HSL, JAK2, and FAT/CD36 [34,47]. FABP4 was found to regulate gene transcription through interaction with JAK2 signalling (Figure 2). FABP4 signalling through JAK2 may induce *SOCS3* gene transcription, resulting in the inhibition of AMPK, increased lipolysis, and reduction of PPARG transcriptional activity [48]. FABP4 retention in the absence of lipolysis and accumulation due to PPARG-induced gene transcription [49] could explain the consecutive increase of PPARG inhibition after long-term transcriptional activation of FAT/CD36. Finally, the reduction of FAT/CD36 levels of expression (Figure 4A) increases basal lipolysis as found in FAT/CD36 deficient adipocytes [30] and after its inhibition in adipose tissue explants (Figure 1A). 

Fatty acid-FAT/CD36 signalling through PPARG for gene transcription and AMPK for lipolysis finely regulates the size of adipocytes. Our results suggest that FABP4 trafficking is a central regulator of the imbalance between reduction versus an increase in size (Figure 9). On the basis of comparative analyses of homogeneous adipocyte maturation (3T3-MBX cell line) versus heterogeneous differentiation allowing to measure the impact of basal lipolysis (3T3-L1 cell lines and human primary cells), we propose that AMPK and PPARG activations are required during early adipogenesis in order to respectively reduce basal lipolysis and increase lipid uptake and storage capacities through transcriptional regulations. The increase of adipocyte size due to dietary fatty acid uptake is associated with an accumulation of intracellular FABP4 which in turn reduces PPARG activity and AMPK activation leading to basal lipolysis. Finally, free fatty acid release and FABP4 secretion improve lipid uptake, and adipocyte size increase in adipocytes with lower size. This hypothesis suggests that FABP4 plays a major role in the regulation of adipocyte size, in order to avoid hypertrophy. The tight regulation of adipocyte size in adipose tissue may explain the stable bimodal distribution of adipocytes according to the cell size as “small” adipocytes (50–70 μm) and “large” adipocytes (70–120 μm) [5,18,35,37].

Finally, induction of adipogenesis in human ASCs by oleic acid was inhibited when the fatty acid was complexed to FABP4 (Figure 5) and inversely regulates the transcription of PPARG-induced adipogenic markers (Figure 8A). Our results are in accordance with in vitro as well as in vivo studies showing that FABP4 interferes with differentiation, downregulates adipogenic markers such as adiponectin and leptin, increases lipolysis in adipocytes, and inhibits differentiation of ASCs at the intracellular level [50,51]. 

FABP4 might enter into cells through endocytosis and then induce intracellular signalling, in addition to intracellular FABP4 accumulation by synthesis, in adipocyte progenitors and mature adipocytes. Thus, FABP4 seems to constitute a central regulator of fat mass expandability.

## 4. Material & Methods

### 4.1. Preparation of Explants, Cell Cultures, and Treatments

Rat epididymal adipose tissue explants were obtained as previously described [23]. Human ASCs were obtained from an anonymous donor (non-diabetic female, 38 years old, 1.73 m height, 98 kg, BMI 32.1; Declaration to French Research Ministry DC n°2008162, Labskin, Lyon, France). The 3T3-L1 and 3T3-MBX sub-clone cell lines (Sigma-Aldrich, Saint-Quentin-en-Yvelines, France), and human ASCs were grown in Dulbecco’s Modified Eagle’s Medium (DMEM) with glucose 4.5 g/L (HG), foetal calf serum (FCS) 10% containing antibiotics and basic Fibroblast Growth Factor (bFGF 20 ng/mL, ThermoFisher Scientific, Invitrogene, Illkirch, France) for ASCs. The adipogenesis procedure and the list of drugs have previously been described [12,23]. Briefly, cells were plated using Scepter counter^®^ (Millipore, Burlington, MA, USA) at 5000 cells/cm^2^. At confluency differentiation was induced in growth HG culture media containing antibiotics and differentiation cocktails 1 (rosiglitazone 20 μM, IBMX 1 mM, dexamethazone 10 μM, insulin 0.05 U/mL,) over 24 h then differentiation cocktail 2 (rosiglitazone, insulin) over 24–48 h, then insulin until the time of the experiment. 

At the time of treatment, an oleic acid stock solution (400 μM complexed to lipid-free bovine serum albumin BSA 10%) was diluted in cell culture media with either 1% BSA (mouse cell cultures) or human lipid-free albumin (hSA) for human cell cultures then incubated at 42 °C over 2 h. Cell cultures were treated with 1 volume of 2× inhibitor media with 1% BSA and then 1 volume of 2× oleic acid solution in order to avoid fatty acid toxicity. AICAR, ATGListatin, BSA, human free fatty acid albumin, dexamethasone, human recombinant FABP4, FABP4 inhibitor, GW6742, GW9662, IBMX, oleic acid, rosiglitazone, and human albumin were purchased by Sigma Aldrich, insulin Actrapid by Novo Nordisk A/S, Danemark), and AP5258 by Clinigenetics (Lyon, France)

### 4.2. Size Determination, Triglyceride, and FAT/CD36 Content Analyses

The methods were previously described in detail [12,23]. Mouse adipose cell lines 3T3-L1 and sub-clone 3T3-MBX were used to analyze either heterogeneous (allowing detection of basal lipolysis) or homogeneous (90–100% differentiation efficiency) cell cultures, respectively. Briefly, cell cultures were analyzed in real-time experiments. XCelligence sensor (ACEA, Agilent Technologies, Les Ulis, France) detects modifications of adipocyte adhesion force related to TG accumulation (reduction of cell index) or release (an increase of cell index). Realtime cell imaging was performed on Cytation 3 plateform (Biotek Instruments, Winooski, VT, USA) [12,23]. Image and fluorescence data were retrieved using the software Gen5 2.08 (Biotek Instruments).

In co-cultures, 3T3-MBX adipocytes were differentiated in either 6-well (Millipore, Sigma Aldrich) or E-plate 96 wells inserts (ACEA) by treatment with oleic acid 10 μM during 3 days before co-culture with 3T3-L1 fibroblasts (env. 80% confluency in low glucose culture media). 

Analyses of size distribution were performed using a Multisizer (Beckman Coulter, Villepinte, France) with adipocytes fixed in formalin 3% after cell detachment by trypsin 0.05% (Sigma Aldrich, Saint-Quentin-Fallavier, France). Triglyceride (TG) contents were quantified in either cell cultures or adipose tissue explants using a cell permeable AdipoRed fluorescent marker (Lonza, Ozyme, Montigny Le Bretonneux, France) on 96-wells plate cultures in real-time experiments performed at the end of treatments. 

Effects of treatments on FAT/CD36 expression, droplet number, droplet size, and cell number, were analyzed by automated quantification of fluorescence after fixation in paraformaldehyde 3 % before labeling with a specific anti-mouse APC-coupled antibody directed against the extracellular domain of FAT/CD36 (2.5 μg/mL, Texas Red filter, Ex 650/Em 661, Biolegend, Ozyme) and/or AdipoRed for TG (FITC filter, Excitation 485/Emission 528) without permeabilization, then Hoechst 33258 in cells permeabilized with 0.1% triton (Sigma Aldrich, Dapi filter, Ex. 355/Em. 465) as previously described [23]. In experiments without detection of FAT/CD36 cells were fixed with formalin 3%. Lipid droplets and nuclei counts were determined for each wavelength according to the controls on images obtained at objective ×4 in 6 to 12 biological replicates. 

Analyses of droplet volume and number per cell (i.e., droplet number normalized to Hoechst counts) were performed on ×4 images. TG content (volume) per cell is the result of the mean number of droplets per cell × mean droplet volume. In each experiment, imaging parameters (i.e., led intensity, camera gain, threshold) were optimized on controls and applied to each sample. In cell cultures, mature adipocytes were considered for diameters up to 50 μm and were analyzed according to mean droplet volume and frequency as previously described [23]. Extracellular FAT/CD36 was quantified by APC intensity detection (normalized to nuclei counts on ×4 images). Realtime AdipoRed uptake by explants was performed by fluorescence intensity measurement after 40 min normalized to the time of AdipoRed treatment on Cytation 3 plateform at 37 °C with 5% CO_2_.

### 4.3. Analysis of FABP4 Binding on 3T3-MBX Cells

Human recombinant FABP4 (Sigma Aldrich) was labeled using Lightning-Link Rapid Alexa Fluor Atto488 Labeling Kit (Ozyme) and loaded onto 3T3-MTX cells plated onto Ibidi dishes (Clinisciences, Nanterre, France) at 100 ng/mL then rinsed in PBS, fixed with formalin 3% then stained with TRITC Phalloïdin 10 nM (Sigma Aldrich) and Dapi 10 ng/mL (Cayman Chemical, Montigny-le-Bretonneux, France) in PBS 0.1% Triton before analysis using the 3D Nanolive microscope (Lausanne, Switzerland).

### 4.4. Gene Transcription Analyzes

Gene transcription analyses were performed by qRT-PCR after total RNA extraction with Trizol (Sigma Aldrich), reverse transcription using Superscript II on 500 μg total mRNA, and quantified with SYBRGreen kit (Roche Diagnosis, Meylan, France) as delta Ct (cycle threshold) as previously described [1,2]. Results were normalized to hypoxanthine guanine phosphoribosyl transferase (HPRT) standard gene quantification and performed in three to four biological replicates. The list of primers has been previously published [12,23].

### 4.5. Bioinformatic Gene Dataset Analyses

The method used for human gene dataset analyses was previously described [12,52,53]. Briefly, human gene datasets were retrieved from either Gene Expression Omnibus (GEO) datasets or published experiments leading to the identification of lists of human genes regulatable according to adipose phenotype (adipose stem cells ASCs, differentiating adipocytes dA, isolated adipocytes or tissues, AT), extracellular signals, intracellular signalling pathways or transcription regulators. Signalling pathways enriched in fatty acids and insulin human gene datasets are reported in Berger & Géloën 2022 [23], those of glucose have been previously published [53]. Mouse FAT/CD36 knock-out gene datasets were retrieved from Sabaouni et al. [54], and that of intracellular FABP4 from Hurley et al., 2012 [55].

### 4.6. Statistics

Experimental data were analyzed on representative experiments on three (12-wells plates) to 6–10 biological replicates (E96-wells plates). In experiments using inhibitors, which were performed in independent experiments, fold changes to corresponding controls were prefered to absolute values, and statistically significant differences were considered for Student *t*-test *p*-values *p* < 0.05. Multiple conditions were analyzed with StatView 4.5 software through an ANOVA test followed by Fisher’s protected least significance difference, post hoc test. ANOVA significantly different values *p* < 0.05 are reported as letters. In bio-informatic data analyses, significant enrichments were considered for Z-test confidence levels > 95% using Epitools software (https://epitools.ausvet.com.au, accessed 3 February 2020, Ausvet Europe). 

## 5. Conclusions

Regulation of adipocyte size requires (1) increased fatty acid storage through lipid uptake and (2) high PPARG transcriptional activity for the synthesis of major pro-adipogenic markers FAT/CD36 and FABP4. A finely tuned management of lipid uptake, storage, and release has been found to be orchestrated by the regulation of PPARG transcriptional activity and AMPK mediated antilipolytic activity through a finely tuned regulation of FAT/CD36 signalling by FABP4 (Figure 9). Although FABP4 is required for adipogenesis, its accumulation by either retention in non-lipolytic adipocytes or uptake in adipogenic precursors may induce a negative control of respectively, size increase resulting in homogenization of adipocyte size, and fatty acid-induced adipogenesis involved in the recruitment of new adipocytes. This hypothesis is in accordance with the observation that FABP4 lack is associated with increased body weight, as well as an increase in body fat without changes in glucose concentration or lipid homeostasis [56]. Thus, FABP4 is a major controller of fatty acid-induced FAT/CD36 mediated regulation of adipose tissue homeostasis by FAT/CD36 signalling.

## Figures and Tables

**Figure 1 ijms-24-01032-f001:**
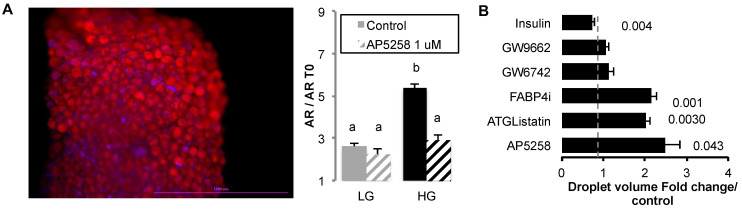
Differential effects of FAT/CD36 inhibition in 3T3-L1 cell cultures versus adipose tissue explants. (**A**) Rat adipose tissue explants were treated during 48 h in low LG (1 g/L) versus high glucose (HG 4.5 g/L) media either alone or in presence of FAT/CD36 inhibitor AP5258 (1 μM). An example of rat adipose tissue explant labeled with AdipoRed (red) and Hoechst 33258 (blue) is presented, the scale bar represents 1 mm. Triglyceride contents were quantified using AdipoRed (AR) uptake after 40 min, fluorescence intensity was normalized at the time the marker was added (T0). Inhibition of FAT/CD36 induced a reduction of lipid content increase observed in HG media by comparison to LG media. Data are presented as mean fold changes AR after 40 min to T0 +/− SEM (n = 8), different letters represent significant differences (ANOVA test, *p* < 0.05). (**B**) Analysis of droplet size regulation 3 days after treatment with oleic acid (10 μM) in HG culture media in 3T3-L1 adipocytes. In presence of either lipolysis inhibitor ATGListatin (1 μM), AP5258 (1 μM), or cell-permeable FABP4 inhibitor (FABP4i 20 μM), lipid droplet sizes were increased. PPARA or PPARG (inhibited respectively by GW6471 and GW9662, 10 μM each) did not affect both cell lines. Insulin (0.05 U/mL) was used as a control of the inhibitory effect on droplet size increase. Results were obtained in independent experiments and normalized to corresponding control media (n = 8 biological replicates), and significant Student *t*-test *p*-values (*p* < 0.05) are indicated.

**Figure 2 ijms-24-01032-f002:**
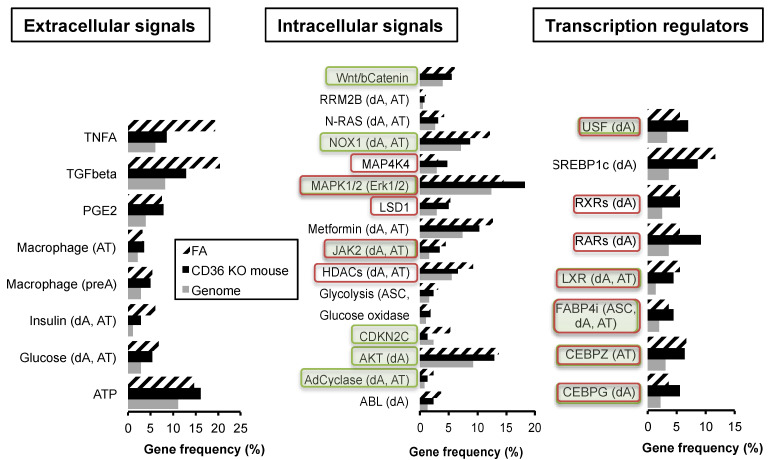
Transcriptional regulation pathways commonly regulated in FAT/CD36 knock-out mice and by fatty acids. Pathways over-represented in adipose cell lineage are reported as ASC (adipose stem cells), AT (adipose tissue), and dA (in vitro differentiated adipocytes). Pathways significantly over-represented in the gene dataset regulated by intracellular FABP4i are indicated in green, in that of extracellular FABP4 in red. Only significantly over-represented pathways in comparison to the genome dataset are reported (z-score confidence levels > 95%).

**Figure 3 ijms-24-01032-f003:**
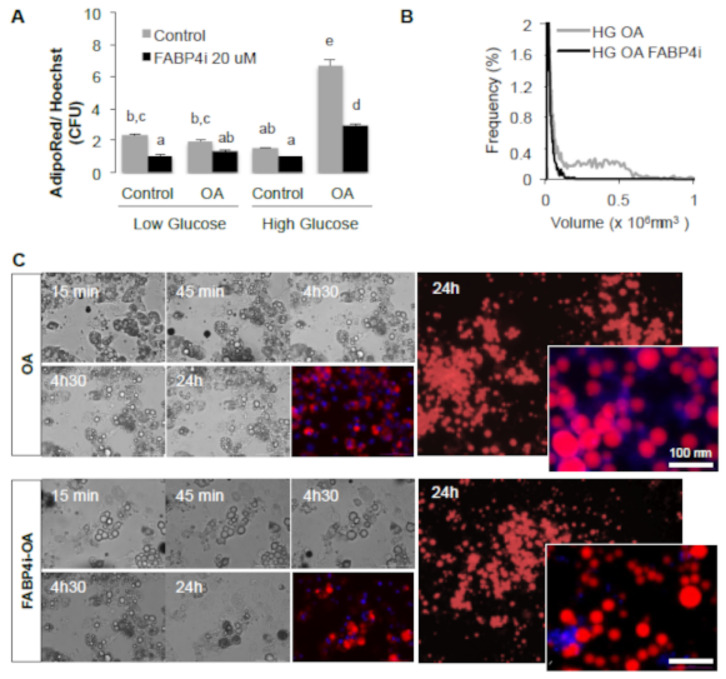
FABP4 is required to induce droplet size increase. (**A**). 3T3-MBX adipocytes treated during 24 h in low glucose (LG, 1 g/L), high glucose (HG, 4.5 g/L) with or without OA (10 μM) and or inhibitor of FABP4 (FABP4i 20 μM) then analyzed by image quantification of Adipored and Hoechst (n = 8 biological replicates). (**B**) Multisizer cell size distribution validates the inhibitory effect of FABP4i on cell size increase (n = at least 5000 cells). Significant differences are represented by letters (ANOVA test *p*-values *p* < 0.05). (**C**) Real-time imaging in high glucose media then merged AdipoRed (red) and Hoechst 33258 (blue), left panel, and representative images at ×4 and ×20 (right panel).

**Figure 4 ijms-24-01032-f004:**
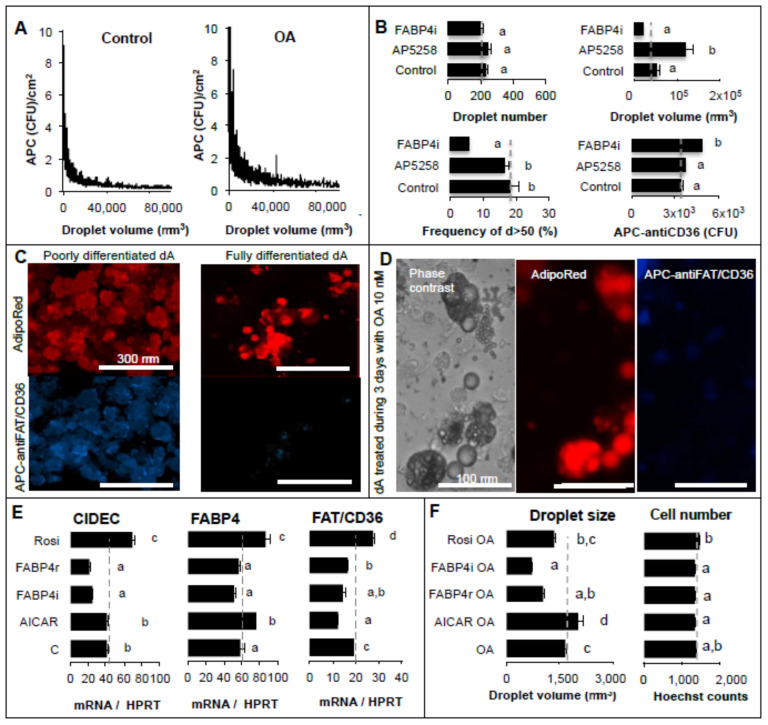
FAT/CD36 expression and activity in adipocytes. (**A**–**D**) The active form of FAT/CD36 was detected using an APC-coupled antibody directed against the extracellular part of the protein (blue color). Lipid droplets were labeled with AdipoRed (red color) and analyses were performed on 3T3-L1 adipocytes 3 days after treatment with oleic acid 10 μM: (**A**) Distribution of extracellular FAT/CD36 co-localized with lipid droplets (env. 5000 counts) is inversely regulated with droplet size. (**B**) Inhibitors of FAT/CD36 (AP5258 1 μM) and FABP4 (FABP4i 20 μM) did not affect droplet number (upper left panel), inhibition of FAT/CD36 promoted the increase of droplet size (upper right panel) although inhibition of FABP4 reduced the frequency of droplets with size up to 50 μm (lower left panel) and increased extracellular FAT/CD36 (lower right panel). (**C**) AdipoRed and APC-FAT/CD36 antibody detection on poorly vs. highly differentiated adipocytes and (**D**) in OA-treated adipocytes. (**E**) Gene transcription analysis of 3T3-MBX adipocytes treated over 3 days with oleic acid 5 μM in order to induce adipogenic genes, then 48 h, with either recombinant FABP4r (20 ng/mL), an inhibitor of FABP4 (FABP4i 20 μM), AMPK activator (AICAR 1 mM) or PPARG activator (rosiglitazone (20 μM). The effects on gene transcription for *CIDEC*, *FABP4*, and *FAT*/CD36 were normalized to the standard *HPRT* gene. Results are presented as mean +/− SEM: n = 8 biological replicates with 3 images at objective ×4 per sample in (**B**,**F**), mean +/− SD in (E) (n = 4 replicates). (**F**) Then their capacity to increase droplet size (left panel) without significant effect on cell number (right panel) was assessed 3 days after treatment with oleic acid (OA 10 μM). (**E**) Results are presented as mean +/− SEM (n = 8 biological replicates). Different letters represent significant differences (Anova, *p* < 0.05).

**Figure 5 ijms-24-01032-f005:**
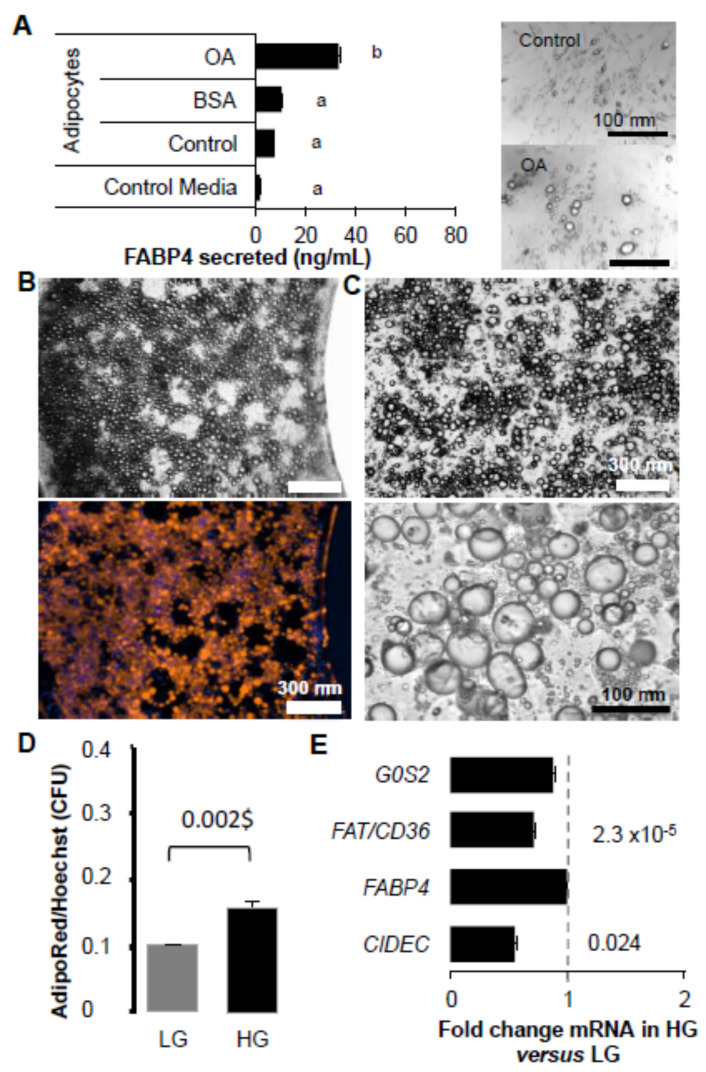
Lipolytic adipocytes inhibit the induction of adipogenesis. (**A**) Human adipocyte maturation was induced by oleic acid 10 μM over 48 h. FABP4 accumulation in the culture media was significantly detected after 48 h. Data are presented as mean FABP4 concentrations +/− SEM (n = 3 biological replicates in 6 wells plates). Different letters represent significant differences (ANOVA test, *p*-values < 0.05). (**B**–**D**) The 3T3-MBX adipocytes were differentiated in either 96-wells (**B**) or 6-wells inserts (**C**), for 4 days then full maturation was induced by treatment with oleic acid 10 μM over 3 days. They were then co-cultured with approx. 80% confluent 3T3-L1 fibroblasts prepared in low glucose media. Co-cultures were performed over 3 days in either low (LG) or high glucose (HG) culture media to induce basal lipolysis in adipocytes. (**B**) Micrographs at ×4 magnification of 96 wells insert adipocytes labeled with AdipoRed (red) and Hoechst (blue). (**C**) Micrographs at objectives ×4 (upper panel) and ×20 (lower panel) of adipocytes grown in 6-well inserts. (**D**) Lipid content in 3T3-L1 fibroblasts was measured as fluorescence intensity fold change of Adipored (AR) normalized to Hoechst (H) in co-cultures vs. alone in 96-wells plates. Data are presented as mean fold changes +/− SEM (n = 7 replicates) with Student’s *t*-test *p*-values *p* < 0.05 in HG vs. LG. (**E**) Transcriptional regulation of adipogenic markers in 3T3-L1 fibroblasts co-cultured with 3T3-MBX adipocytes during 3 days in high vs. low glucose media. Data are presented as mean fold changes mRNA (normalized to standard HPRT gene) of HG versus LG condition +/− SD of 3 replicates (6-wells inserts) with significant Student’s *t*-test *p*-values *p <* 0.05.

**Figure 6 ijms-24-01032-f006:**
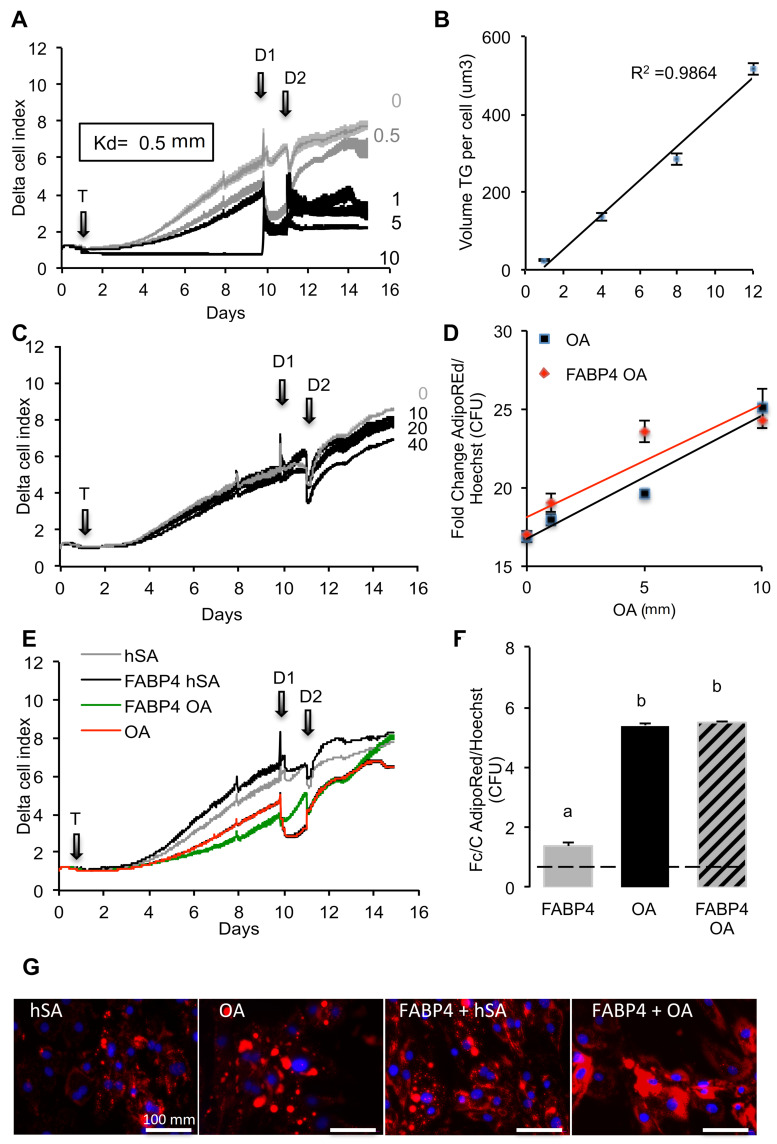
Extracellular **Fatty acid binding protein 4** (FABP4) inhibits oleic acid (OA)-induced adipogenesis in human adipose stem cells (ASCs). At time (T) human ASCs were exposed during proliferation to OA complexed to human albumin (hSA) (**A**,**B**), recombinant FABP4 at several doses (**C**,**D**), or hSA 1%, OA 10 μM and/or FABP4 20 ng/mL (**E**–**G**), then differentiation was induced (D1 and D2). The effect of OA on differentiation was revealed by the dose-dependent effect on cell adhesion force monitored as a cell index normalized at the time cells were loaded. (**A**) Real-time monitoring of cell index (representing adhesion force) shows the dose-dependent promoting effect of oleic acid on adipogenesis, i.e., reduction of cell adhesion force after induction of differentiation for OA doses up to 0.5 μM, (**B**) validated by quantification at the end of the experiment of the mean droplet cell size (droplet count normalized to nuclei counts) on ×4 images. (**C**) Real-time monitoring of adipogenesis in presence of FABP4 at several doses shows no significant effect. (**D**) TG accumulation (AdipoRed fluorescence intensity normalized to cell number) due to OA uptake is independent of OA complexation to hSA versus FABP4. (**E**) Real-time monitoring of adipogenesis shows that although OA complexation to hSA improved adipogenesis (reduction of cell index after induction of differentiation), its complexation to FABP4 inhibited adipogenesis (an increase of cell adhesion after induction of differentiation), (**F**) without affecting accumulation of TG detected as described in (**D**). (**G**) Representative micrographs of merged Adipored (red) and Hoechst 33258 (blue) human cells at the end of experiments. Data are presented as mean +/− SEM and letters represent significant differences (n = 8 biological replicates, ANOVA test, *p* < 0.05) in (**F**).

**Figure 7 ijms-24-01032-f007:**
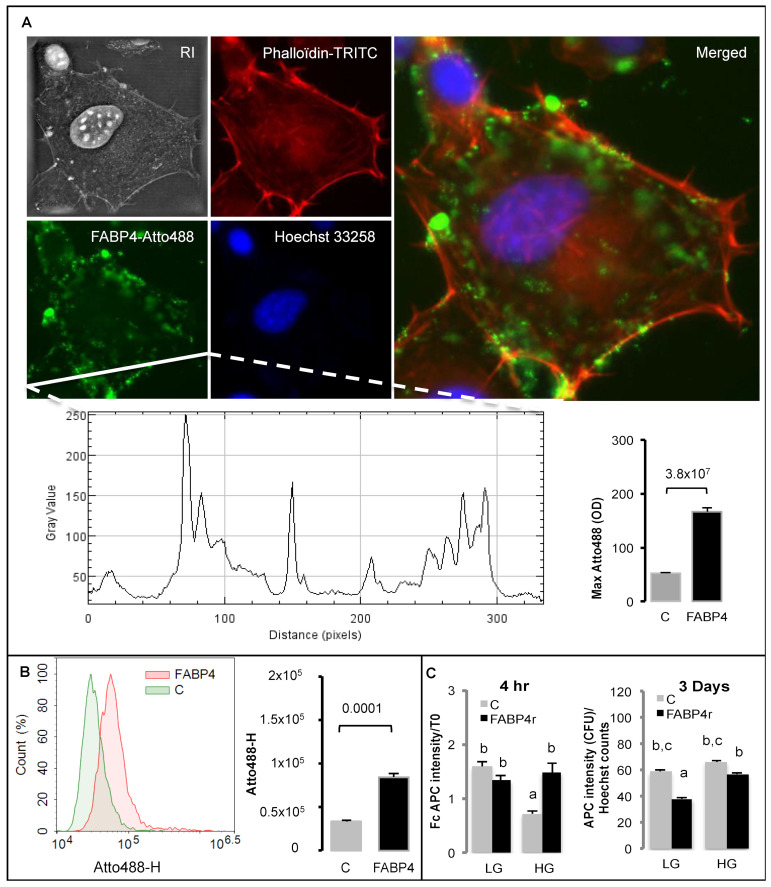
Fatty acid binding protein 4 (FABP4) uptake and regulation of active FAT/CD36 form in 3T3-MBX cells (**A**) Nanolive imaging after 3 h treatment with Atto488-FABP4 (FABP4r 100 ng/mL) and quantification of max Atto488 intensity per cell. (**B**) Atto488-FABP4 quantification by cytometry. (**C**) Effect of FABP4r on extracellular FAT/CD36 detected with APC-coupled anti-FAT/CD36 antibody after 4 h and after 3 days. Data are presented as mean values +/− SEM (n = 10 cells in A, 8 biological replicates in (**B**,**C**)) with significant differences indicated by Student *t*-test *p*-values in (**A**,**B**), and different letters in (**C**) (Anova test *p*-values < 0.05).

**Figure 8 ijms-24-01032-f008:**
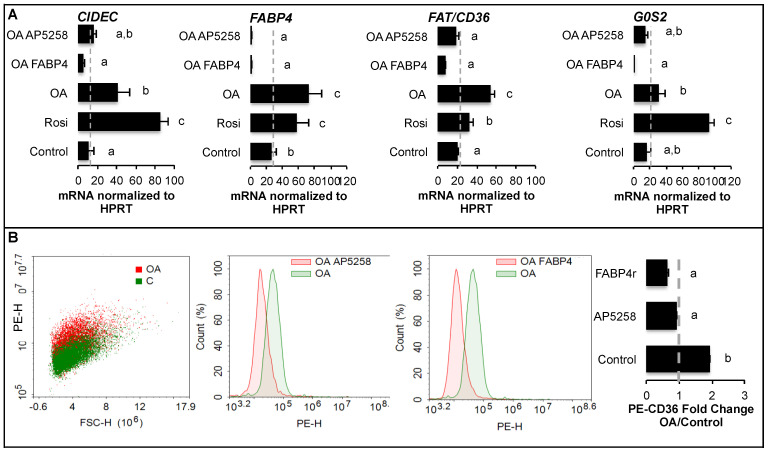
Fatty acid binding protein 4 (FABP4) inhibits the pro-adipogenic programme mediated by fatty acid/FAT/CD36 pathways in 3T3-L1 fibroblasts. Treatments were performed in DMEM high glucose media over 24 h with oleic acid (OA 0.5 μM), recombinant FABP4 (FABP4 20 ng/mL), FAT/CD36 inhibitor (AP5258, 1 μM) or rosiglitazone (Rosi 10 μM). (**A**) Gene transcription analyses by qRT-PCR, normalized to *HPRT* standard, show the inhibitory effect of FABP4 and FAT/CD36 inhibition on their own induction by OA. (**B**) Phycoerythrin (PE)-coupled anti CD36 antibody was analyzed by cytometry and showed that both FABP4r and FAT/CD36 inhibitor reversed the induction of extracellular FAT/CD36 by OA. Data are presented as mean values +/− SEM ((**A**): n = 8 biological replicates, (**B**): 4 replicates; at least 5000 cells/sample) with significant differences represented by letters (ANOVA *p*-values < 0.05).

**Figure 9 ijms-24-01032-f009:**
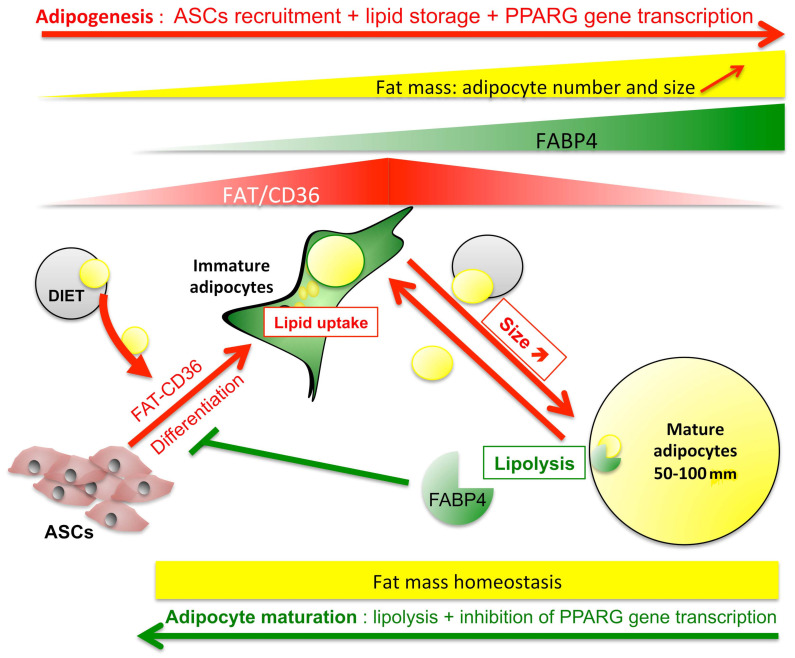
Putative mechanism of Fatty acid binding protein 4 (FABP4) role in the regulation of fat mass. Fat mass regulation proceeds through pro-adipogenic processes allowing lipid storage (red pathways) requiring fatty acid-induced PPARG gene transcription, lipid uptake, and inhibition of lipolysis. Fatty acid interaction with FAT/CD36 controls both AMPK anti-lipolytic activity and PPARG mediated gene transcription, including those of *FAT/CD36* and *FABP4*. FAT/CD36 increases during adipogenesis together with FABP4. Increased accumulation of FABP4 in absence of lipolysis in mature adipocytes is a limiting factor inducing reduction of FAT/CD36 through PPARG inhibition (negative feedback), which in turn alters its antilipolytic activity. Finally, FABP4 exerts a limitation to a fatty acid-induced increase of adipocyte size through the restoration of basal lipolysis (green pathways), then promotes small or immature adipocytes to reach the optimal size but inhibits the recruitment of new adipocytes. The model suggests that under normal conditions, lipid storage is then optimized to obtain a homogenous population of adipocytes (in a range of 50–100 μm) allowing to monitor the metabolic response to diet uptake versus energy expenditure.

## Data Availability

All data are presented in the main text and/or in Appendix A. Gene datasets retrieved from the literature are referenced in either the main text or in the Appendix A.

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
