# Peer review of "FABP4 Controls Fat Mass Expandability (Adipocyte Size and Number) through Inhibition of CD36/SR-B2 Signalling"

_ijms, 2023, doi:10.3390/ijms24021032_

Round 1

Reviewer 1 Report

Dear authors,

In the present work. entitled “FABP4 controls fat mass expandability (adipocyte size and number) through inhibition of CD36/SR-B2 signalling” Berger et al investigated the impact of FABP4 on adipocytes metabolism size and adipogenesis.

The abstract is confusing and needs to be rewritten, what are you referring to with “Lipid storage into triglyceride droplets is imbalanced by lipogenesis (mainly lipid uptake in mature adipocytes and lipolysis) “? Is this referring to uptake or lipogenesis?

In the first sentence of the introduction, there is an issue with micrometers units referring to adiposize.

In the introduction, the authors are also referring to lipogenesis as de novo lipid synthesis, lipid uptake, and lipolysis. Lipid uptake and lipolysis shouldn’t be inserted in the definition of lipogenesis, is this correct? Sentences must be rephrased.

HG abbreviation is introduced but not written in full form. Figure 1A data are referring to something published somewhere else, am I correct? This must be moved into supplemental information as it is a confirmation of previously observed data. The representative image of panel 1A is referring to 1mm as a scale, is this correct? Image scale must be automatically inserted by the program used in the analysis; this seems to be added during image layout. Image quality is very poor, and reports just one genotype must be reported in the image. The graph in figure 1A reports SEM for some points and not all and seems to be moved in respect of the line. How is possible that the acquiring time is 48 hours and graph 1A reports at a maximum of 40 with the axis legends with Time (mn) that I think is referring to min. Letters are used for significative differences but nowhere is explained to what. In the figure legends, authors are referring to TG quantification which is wrong as just Adipored quantification has been used.

Are the HG and LG able to affect also adipocyte size? Authors are modifying substrates providing a higher concentration of glucose and saying that this is promoting lipogenesis or lipid uptake, this is impossible to say, and that is independent of basal lipolysis. What does this mean? What is the reason of providing data about LG and HG? Increased LD size following FABP4, CD36, and ATGL inhibition is suggesting an implication for LD size, is this happening also with HG? This will shed the light on lipogenesis. 

In line 133 FAT/CD35 is wrongly written

In figure 2 isgene dataset of mice  compared to human genome and cells treated with FA? What is treated with FA is not even reported but Is this comparison possible? I would rather compare cells alone with CD36 inhibitors and with FA. 

ASCs are somewhere reported as hAS, is this correct? These mistakes in reporting names, axis legends and information, coupled with poor graphical representation make comprehension of the text almost impossible. I’m not even sure on what the authors mean with lipolysis in the text

Is Panel E of figure 5 reporting a fold of change? This should be reported

What is the meaning of co-cultured adipocytes with 3T3-L1 fibroblasts? This is not reported anywhere.

Author Response

In the present work. entitled “FABP4 controls fat mass expandability (adipocyte size and number) through inhibition of CD36/SR-B2 signalling” Berger et al investigated the impact of FABP4 on adipocytes metabolism size and adipogenesis.

The abstract is confusing and needs to be rewritten, what are you referring to with “Lipid storage into triglyceride droplets is imbalanced by lipogenesis (mainly lipid uptake in mature adipocytes and lipolysis) “? Is this referring to uptake or lipogenesis?

The sentence has been rewritten:

 "In mature adipocytes lipid accumulation as triglycerides into droplets is imbalanced by lipogenesis (mainly lipid uptake and lipolysis). In previous studies we showed that adipogenesis induced by oleic acid is signed by size increase and reduction of FAT/CD36 (SR-B2) activity [1] [2]."

In the first sentence of the introduction, there is an issue with micrometers units referring to adiposize.

In the introduction, the authors are also referring to lipogenesis as de novo lipid synthesis, lipid uptake, and lipolysis. Lipid uptake and lipolysis shouldn’t be inserted in the definition of lipogenesis, is this correct? Sentences must be rephrased.

Thes sentence have been modified:

L55  

"Fat storage in adipocytes is the result of 1) free fatty acid (FFA) uptake and storage as triglycerdies (TG) into droplets, 2) free fatty acid release by lipolysis [13] [12]; 3) induction of intracellular signaling pathways including fatty acid receptors.."

L63

"Adipogenesis is induced and regulated by lipogenesis, including de novo lipid synthesis and lipid uptake as well as lipolysis".

HG abbreviation is introduced but not written in full form.

The sentence has been modified:

L110: "better efficiency in high (4.5 g/L, HG) versus low glucose (1 g/L, LG) media"

Figure 1A data are referring to something published somewhere else, am I correct?

This must be moved into supplemental information as it is a confirmation of previously observed data.

Figure 1A: in a previous study (Berger and Géloën 2022) rat explants were treated during 48 h with oleic acid in either low or high glucose culture media, and adipocyte size was monitored using AdipoRed size counts on images.

In the present figure the rat explants were treated also during 48 h in either low or high glucose culture media in order to test the effect of FAT/CD36 inhibitor AP5258 on adipose lipid contents.

The representative image of panel 1A is referring to 1mm as a scale, is this correct? Image scale must be automatically inserted by the program used in the analysis; this seems to be added during image layout. Image quality is very poor, and reports just one genotype must be reported in the image.

The image of explant has been increased to improve the quality as muc as possible. The scale bar (1 mm) inserted by the software is shown but the resolution of scale bar is poorly detected. The effects were not apparent in images, therefore only a representative control explant is presented.

The graph in figure 1A reports SEM for some points and not all and seems to be moved in respect of the line. How is possible that the acquiring time is 48 hours and graph 1A reports at a maximum of 40 with the axis legends with Time (mn) that I think is referring to minSince the Effect on size could not be detected, (as previously described with OA treatments), therefore the TG content was assessed at the end of experiment, by treatment with AdipoRed during 40 min, the fluorescence intensity was normalized to time the marker was added (T0). In order to clarifiy the experiment, the time-dependent responses to AdipoRed has been removed.

. In the figure legends, authors are referring to TG quantification which is wrong as just Adipored quantification has been used.

The legend has been modified

Triglyceride contents were quantified using AdipoRed (AR) uptake after 40 min, fluorescence intensity was normalized at time of the marker was added (T0). Inhibition of FAT/CD36 induced a reduction of lipid content increase observed in HG media by comparison to LG media.

. Letters are used for significative differences but nowhere is explained to what

Data are presented as mean fold change AR after 40 min to T0 +/-SEM (n=8), different letters represent significant differences (Anova test, p-<0.05).

Are the HG and LG able to affect also adipocyte size? Authors are modifying substrates providing a higher concentration of glucose and saying that this is promoting lipogenesis or lipid uptake, this is impossible to say, and that is independent of basal lipolysis. What does this mean? What is the reason of providing data about LG and HG?

The controls suggest that in HG media, the capacity of AdipoRed uptake was higher thant in LG, suggesting that after 48h de novo lipogenesis could occur. In the concern of Fig.1A the result section has been rewritten:

In a previous study, we showed using AdipoRed probe incorporation that oleic acid uptake induced size increase with better efficiency in high (4.5 g/L, HG) versus low glucose (1 g/L, LG) media [2]. In the present study rat explants were treated during 48h in either LG or HG media and/or optimized concentration of FAT/CD36 inhibitor AP5258 (Figure 1A). Triglyceride (TG) contents were then assessed using the fluorescent probe AdipoRed incorporation in 40 min. We found that in HG media Adipored incorporation was higher in HG than in LG media, suggesting that fatty acid de novo synthesis from glucose was detected in these experiments. AP5258 inhibited this effect, suggesting that FAT/CD36 plays a major role in the regulation of TG synthesis in adipose tissues.

Increased LD size following FABP4, CD36, and ATGL inhibition is suggesting an implication for LD size, is this happening also with HG? This will shed the light on lipogenesis. 

In Fig. 1B experiments were performed in HG culture media with OA in order to study the mechanism involved after lipid uptake. De novo fatty acid synthesis is mainly involved in the pro-adipogenic process induced by insulin in immature cells. Using fatty acid synthase inhibitors, such as irgasan, the effect of OA on adipocyte size increase was even increased (unpublished observations).

Fig1 The legend has been modified :

B- Analysis of droplet size 3 days after treatment with oleic acid (10 mM) in HG culture media in 3T3-L1 adipocytes.

together with the main text

L 140: "3T3-L1 adipocytes were treated with low dose of oleic acid in HG culture media in presence of several inhibitors at optimized concentrations during 3 days."

In line 133 FAT/CD35 is wrongly written

corrected

In figure 2 isgene dataset of mice  compared to human genome and cells treated with FA? What is treated with FA is not even reported but Is this comparison possible? I would rather compare cells alone with CD36 inhibitors and with FA. 

In figure 2 FAT/CD36 ko and fatty acid datasets were compared independantly to human genome rather than to each other because such an approach gave more informations as described in the figure, such as relevance to adipose phenotype and commonly enrichment in pathways over-represented in FABP4 datasets.

The method hase been validated in several other publications not cited in the present article:

Berger et al 2011 doi:10.1152/physiolgenomics.00013.2010. (human cancer cells response to adiponectin)

Berger et al 2015 http://dx.doi.org/10.1155/2015/821761 (liver phenotype and hepatocellular carcinoma cells)

Berger et al 2017 doi: 10.3390/ijms18071573. (intestinal and colon cancer phenotypes, fatty acid signalling in cancer phenotypes)

L160 a sentence has been added

[2]. Although the datasets are obtained from experiments performed in different conditions and on different kind of samples, previous studies have shown that significant enrichments in gene datasets allow identification of signalling pathway crosstalks.

The fatty acid dataset was described in a previous study (Berger et al, 2012 doi: 10.1002/biot.201200188). The reference has been added in Suppl file 3:

ASCs are somewhere reported as hAS, is this correct?

ASC has been check to represent Adipose stem cells in all the text.

hSA was used as the abbreviation for human serum albumin used in some experiments instead of BSA (bovine serum albumin), which was found to induce toxicity in several experiments on human cells.

These mistakes in reporting names, axis legends and information, coupled with poor graphical representation make comprehension of the text almost impossible. I’m not even sure on what the authors mean with lipolysis in the text

The text has been extensively corrected at the level of typo errors and english

Efforts have been made to improve the graphical quality of figures.

Is Panel E of figure 5 reporting a fold of change? This should be reported

Figure 5E: the figure x axis has been modified with the corresponding paramater: Fold change mRNA in HG versus LG. Legend was also corrected

What is the meaning of co-cultured adipocytes with 3T3-L1 fibroblasts? This is not reported anywhere.

L212 a senetnce has been added to explain the use of co-cultures:

In a previous study, we found that basal lipolysis can be detected in high glucose culture media of highly differentiated adipocytes with heterogeneous sizes [2]. We used fully differentiated 3T3-MBX adipocytes obtained by pre-treatment with OA during 3 days (Fig. 5B and 5C) in order to study in co-cultures how basal lipolysis and FABP4 release can modulate the recruitment of new adipocytes using 3T3-L1 fibroblasts.

Reviewer 2 Report

Obesity-related diseases have become major healthcare burdens, hence such research is highly desired. I believe the performed research adequately explains the dichotomy of adipocyte size at the molecular level, shedding light at potential novel molecular targets of intervention in parallel.

AdipoRed fluorescent signals are shown in green color (as stated by figure caption). I believe it would make it easier to comprehend the results if they were shown in red color, simply by swapping RGB channels in e.g. CorelDraw without restaining any samples.

The presentation of the reults is of good overall quality with small minor issues. Special characters (eg. micrometer etc) have become illegible throughout the manuscript in the accessible pdf format. This might be due to conversion issues during processing, please double check. Also, there are occasional typing errors and French-spelled words that shall be easily identified and corrected using standard Word grammar option if set to English. 

Author Response

Obesity-related diseases have become major healthcare burdens, hence such research is highly desired. I believe the performed research adequately explains the dichotomy of adipocyte size at the molecular level, shedding light at potential novel molecular targets of intervention in parallel.

AdipoRed fluorescent signals are shown in green color (as stated by figure caption). I believe it would make it easier to comprehend the results if they were shown in red color, simply by swapping RGB channels in e.g. CorelDraw without restaining any samples.

AdipoRed in red color: every figure containing AdipoRed images have been transformed with red color for AdipoRed and blue color for nuclei label.

The presentation of the reults is of good overall quality with small minor issues. Special characters (eg. micrometer etc) have become illegible throughout the manuscript in the accessible pdf format. This might be due to conversion issues during processing, please double check. Also, there are occasional typing errors and French-spelled words that shall be easily identified and corrected using standard Word grammar option if set to English.

Quality of figures has been improved as much as possible

Manuscript has been carefully checked for English and typo errors

Round 2

Reviewer 1 Report

Dear Authors,

thank you for your revision, the paper has improved and the readability also has. the concept of lipolysis and de novo sinthesis has now been clarified as well as the use of fibroblast. The figure 9 still have some proplem with format of the letterst that in the pdf are reported as question mark, plese check carefully.

Author Response

RESPONSE TO REFEREE 2

Open Review

English language and style

( ) English very difficult to understand/incomprehensible
( ) Extensive editing of English language and style required
( ) Moderate English changes required
(x) English language and style are fine/minor spell check required
( ) I don't feel qualified to judge about the English language and style

The text has been extensively read again and numerous typo errors have been corrected as well minor style and/ or language modifications have been checked. It should me noticed that nulerous symbols have been modified in the edited pdf version, their have been corrected and indicated in red in the text.

Yes

Can be improved

Must be improved

Not applicable

Does the introduction provide sufficient background and include all relevant references?

( )

(x)

( )

( )

Are all the cited references relevant to the research?

( )

(x)

( )

( )

Is the research design appropriate?

( )

(x)

( )

( )

Are the methods adequately described?

( )

(x)

( )

( )

Are the results clearly presented?

( )

(x)

( )

( )

Are the conclusions supported by the results?

( )

(x)

( )

( )

Comments and Suggestions for Authors

Dear Authors,

thank you for your revision, the paper has improved and the readability also has. the concept of lipolysis and de novo sinthesis has now been clarified as well as the use of fibroblast. The figure 9 still have some proplem with format of the letterst that in the pdf are reported as question mark, plese check carefully.

The quality of all figures has been improved as much as possible, the figure 9 has been submitted in two different formats to help editing.

Submission Date

08 December 2022

Date of this review

27 Dec 2022 11:02:11
